# Genome-Wide Association Study of Seed Morphology Traits in Senegalese Sorghum Cultivars

**DOI:** 10.3390/plants12122344

**Published:** 2023-06-16

**Authors:** Ezekiel Ahn, Jacob Botkin, Vishnutej Ellur, Yoonjung Lee, Kabita Poudel, Louis K. Prom, Clint Magill

**Affiliations:** 1USDA-ARS Plant Science Research Unit, St. Paul, MN 55108, USA; 2Department of Plant Pathology, University of Minnesota, Saint Paul, MN 55108, USA; botki009@umn.edu (J.B.); lee03479@umn.edu (Y.L.); 3Molecular Plant Sciences, Washington State University, Pullman, WA 99164, USA; 4Department of Agronomy and Plant Genetics, University of Minnesota, St. Paul, MN 55108, USA; poude024@umn.edu; 5USDA-ARS Southern Plains Agricultural Research Center, College Station, TX 77845, USA; louis.prom@usda.gov; 6Department of Plant Pathology and Microbiology, Texas A&M University, College Station, TX 77843, USA

**Keywords:** sorghum, seed, morphology, area size, seed color, circularity, GWAS

## Abstract

Sorghum is considered the fifth most important crop in the world. Despite the potential value of Senegalese germplasm for various traits, such as resistance to fungal diseases, there is limited information on the study of sorghum seed morphology. In this study, 162 Senegalese germplasms were evaluated for seed area size, length, width, length-to-width ratio, perimeter, circularity, the distance between the intersection of length & width (IS) and center of gravity (CG), and seed darkness and brightness by scanning and analyzing morphology-related traits with SmartGrain software at the USDA-ARS Plant Science Research Unit. Correlations between seed morphology-related traits and traits associated with anthracnose and head smut resistance were analyzed. Lastly, genome-wide association studies were performed on phenotypic data collected from over 16,000 seeds and 193,727 publicly available single nucleotide polymorphisms (SNPs). Several significant SNPs were found and mapped to the reference sorghum genome to uncover multiple candidate genes potentially associated with seed morphology. The results indicate clear correlations among seed morphology-related traits and potential associations between seed morphology and the defense response of sorghum. GWAS analysis listed candidate genes associated with seed morphologies that can be used for sorghum breeding in the future.

## 1. Introduction

Sorghum [*Sorghum bicolor* (L.) Moench] is a gluten-free cereal widely consumed throughout Africa [1,2]. It is a climate-resilient and drought-tolerant crop used for animal feed, biofuels, forage, ethanol production, and fodder preservation; it is arguably one of Africa’s most versatile food crops [3]. Plant genetics researchers have frequently employed community association panels to investigate the genetic basis of naturally occurring phenotypic variation in several traits [4]. Sorghum germplasm lines from West and Central Africa are cultivated in rainy and high-humidity regions and have been a significant source of critical agronomic traits such as fungal disease resistance [5]. To date, Senegalese germplasms have been extensively tested through genome-wide association studies (GWAS) to identify novel sources of resistance genes against fungal diseases such as anthracnose and head smut [5,6,7], but the germplasms have yet to be widely studied for other agronomically important traits such as seed morphology.

Morphological variations in seed characters, such as differences in seed size and shape, are important traits in plant identification and classification of taxa, and are useful parameters for analyzing plant biodiversity [8,9]. Seed morphology is of agronomic importance, as it reflects genetic, physiological, and ecological components that affect yield, quality, and market price [8]; for instance, acceptance of high lysine-containing sorghums is limited due to many problems associated with their opaque kernel characteristics [10,11]. Breeding targets encompass seed shape and size, as seed size-related traits bear particular importance, and a comprehensive understanding of the genes underlying seed morphology equips breeders with the capacity to develop novel cultivars harboring desired characteristics [12].

Sakamoto et al. evaluated 329 sorghum germplasms from various origins and identified SNPs potentially associated with seed morphology, including SNP loci S01_50413644, S04_59021202, and S05_9112888 based on multi-traits GWAS [13]. Zhang et al. identified 73 quantitative trait loci (QTL) related to grain color and tannin content in Chinese sorghum accessions, and a new recessive allelic variant of Tannin2 was discovered [14]. A GWAS conducted on a diverse set of 635 Ethiopian sorghum accessions found variations in loci harboring seed protein genes involved in seed storage, late embryogenesis, and tannin biosynthesis, all of which are associated with sorghum grain mold resistance [15].

To add to our understanding of sorghum seed morphology, in this study, 162 Senegalese sorghum germplasms along with controls (BTx623, PI609251, and PI659985, which are widely used cultivars in sorghum research) were evaluated for various seed morphologies, including seed area size, length, width, length-to-width ratio (LWR), perimeter, circularity, the distance between the intersection of length and width (IS) and center of gravity (CG), and seed darkness and brightness. In addition to identifying variation for each trait, possible correlations among the characteristics were evaluated across the subset of the Senegalese collection. Furthermore, Ahn et al. previously studied identical germplasms to evaluate resistance to anthracnose at the seedling and 8-leaf stages and for head smut based on distinctive spot appearance rate and average time for spot appearance on the first leaf under water [6,7,16,17]. By taking advantage of the previous studies, the seed morphology-related traits were also analyzed to identify potential correlations with the traits associated with anthracnose and head smut. Finally, the phenotypic data collected from over 16,000 seeds for the traits were combined with 193,727 single nucleotide polymorphisms (SNPs) throughout the genome to perform GWAS regarding single traits and multi-traits. The top candidate SNPs were tracked back to the reference sorghum genome, resulting in the identification of multiple candidate genes potentially associated with seed morphology-related traits.

## 2. Results

### 2.1. Seed Morphologies

Based on the two-tailed test, the ANOVA for the 165 accessions, including BTx623, PI609251, and PI659985, showed significant differences, with *p* < 0.0001 for all evaluated traits (raw data available through Appendix A). The Shapiro–Wilk test identified normal distribution for the distance between IS and CG (*p* = 0.086), but other traits showed abnormal distributions with *p* < 0.0001 (Figure 1). Brightness and circularity were skewed to the right, and the other traits had a few to multiple outliers (Figure 1). The top five cultivars for each trait are shown in Table 1 (detailed phenotypic data are available through Appendix A). For example, the area size for PI514293 was 6.67 ± 0.86 mm^2^, while that for PI514404 was 19.26 ± 3.41 mm^2^ (Table 1 and Figure 2). Similarly, the seed colors between PI514471 and PI514419 showed great contrast (Table 1 and Figure 3). Significant phenotypic variations were observed in other traits across the population as well.

### 2.2. Correlations among the Seed Morphology-Related Traits

Based on Pearson’s correlation analysis, all evaluated seed size-related traits correlate to each other except for perimeter-circularity and LWR-circularity (Figure 4 and Table 2). Seed brightness showed no correlation to all seven seed size-related traits indicating the seed color-related trait is highly independent of seed size-related traits.

A PCA using eight seed morphology-related traits revealed that PC1 and PC2 explain 75.83% of the overall variation (Figure 5). The plot of the partial contribution of variables for eight traits revealed that PC1 comprises area size, perimeter, length, width, and length-to-width ratio (Figure 6). Circularity and distance between IS and CG mainly contribute to PC2. Seed brightness contributes mostly to PC3.

### 2.3. Correlations between the Seed Morphology-Related Traits and Anthracnose and Head Smut Resistance Responses in Sorghum

Based on multivariate correlation studies between seed morphology-related traits and sorghum responses to anthracnose and head smut, we identified two major correlations supported with significant *p*-values. There was a moderately strong negative correlation between head smut spot appearance rate (%) and circularity (Figure 7a). On the other hand, head smut spot appearance rate (%) showed a moderate positive correlation to distance between IS and CG, which are directly associated with seed morphology and circularity (Figure 7b). Although *p*-values were higher than 0.05, seed area size and length were also negatively correlated to the spot appearance rate (*p*-values ≈ 0.05). Other seed morphology-related traits did not show correlation to the diseases’ associated traits.

### 2.4. Population Structure and GWAS Analysis

The population structure analysis revealed that there are two major groups across the accessions tested in this study (Figure 8). The dendrogram displayed nearly identical results to the admixture plot (Figure 9). Overall, the population structure analysis aligned with previous studies indicating that the botanical subrace played a major role in shaping the diversity patterns of the population [7,18,19]. PCA plots, along with phylogenetic trees, were omitted as it has been reported in a recent study in the population [7].

Figure 10 and Table 3 show the top SNPs identified from GWAS and their associated genes. LD heatmaps highlighted LD of regions near the statistically significant SNP loci, indicating low LD (Figure 11). Overall, 100 SNP variants passed the Bonferroni threshold before secondary filtering with a *t*-test. The number of SNPs for each trait varied (3 SNPs for area size, 11 SNPs for perimeter, 0 SNP for length, 9 SNPs for width, 3 SNPs for LWR, 0 SNP for distance between IS and CG, 56 SNPs for PCs and 18 SNPs for brightness). After filtering with a *t*-test, multiple SNPs were excluded from the list, leaving the top SNPs that can be used as novel sources for sorghum seed morphology-related traits in breeding programs.

## 3. Discussion

Seed weight and size are critical yield components, and selecting for large seeds has long been a goal in crop domestication [20,21]. Measures of size and shape in seeds and their correlation are equally essential in current breeding to improve yield or quality [8]. As one of the most important crops worldwide, sorghum seed morphologies and their associations with molecular markers are of potential use not only for breeding but for evaluating the role of specific genes in seed shape and size. This study explored eight important sorghum seed morphology traits in Senegalese sorghum accessions that have yet to be extensively studied regarding seed-related traits. Figure 4 and Table 2 show that many seed size-related traits, including area size, perimeter, length, width, and length-to-width ratio, were correlated. Circularity and distance between IS and CG were also associated with other seed size-related traits. However, based on PCA analysis (Figure 5 and Figure 6), these traits were not closely grouped with other seed traits. Seed brightness, primarily explained by PC3, was not significantly associated with the other seed traits.

In the multivariate correlation studies examining the relationship between seed morphology-related traits and sorghum responses to anthracnose and head smut, we observed two significant correlations. Firstly, we found a moderate negative correlation between the head smut spot appearance rate (%) and circularity. Secondly, the head smut spot appearance rate (%) showed a moderate positive correlation with the distance between IS and CG. Craig and Frederiksen conducted a seedling inoculation for sorghum using peat pellets against *Sporisorium reilianum* (Kühn) Langdon & Fullerton (syns. *Sphacelotheca reiliana* (Kühn) G.P. Clinton and *Sorosporium reilianum* (Kühn) McAlpine) causing sorghum head smut [16,22]. The sorghum seedlings at the 1-leaf stage were inoculated with teliospore cultures. After four days, the seedlings were submerged in water-filled test tubes, and the presence of brown or dark spots on the first leaf blade distinguished susceptible genotypes from resistant ones. Craig and Frederiksen explained the spots caused by the fungal pathogen, but it is unclear if the spots are present due to fungal infection or due to a plant defense response [16,22]. Although the observed correlations may be mere coincidences, Seiwa et al. reported results suggesting that seed size may play a role in conspecific negative distance-dependent seedling mortality and negative density-dependent seedling survival variation (CNDD), and that seed size may promote species coexistence by influencing distance-dependent pathogen attacks, especially those related to leaf diseases in eight tree species [23]. A study conducted in the Peruvian Amazon suggested a positive correlation between tree seed size and susceptibility to pathogen attack [24]. Specifically, the results indicated that larger and shade-tolerant seeds exhibited a higher vulnerability to pathogen attack than smaller seeds relying on light dependence [24]. A positive relationship between seed weight and susceptibility to pathogens was also found [24]. Ahn et al. [7] reported top candidate genes associated with the spot appearance when inoculated with *S. reilianum* at the seedling stage. The top candidate genes (F-box and leucine-rich repeat protein (Sobic.004G202700), ankyrin repeats (Sobic.002G174700) and xyloglucan endotransglucosylase (Sobic.004G273200)) were located near growth and development-related genes such as rhodanese-like domain-containing protein-like (Sobic.004G202600) [25], cellulase/endoglucanase (Sobic.004G202800) [26], protein kinase AFC1 (Sobic.004G202500) [27], serine/threonine protein kinase (Sobic.002G174801) [28], legume lectin domain (Sobic.002G174600) [29], transcription factor jumonji (jmj) family protein/zinc finger (C5HC2 type) family protein (Sobic.004G273100) [30], ubiquitin and ubiquitin-like proteins (Sobic.004G273300) [31], and MYB-like DNA-binding domain (Sobic.004G273000) [32]. Therefore, it is speculated that the correlations between the head smut spot appearance rate and the two morphology-related traits are due to strong genetic linkage.

Zinc finger proteins play essential roles in plant growth, development, and responses to abiotic stresses such as drought, salt, temperature, reactive oxygen species, and harmful metals [33]. As listed in Table 3, zinc finger-associated genes were linked with multiple SNPs potentially associated with seeds (SNP loci for area: S06_12058855, perimeter: S02_36482136 and S07_16244875, width: S06_12058855 and S07_16244901, and PCs: S06_54692844 and S06_54708681). MicroRNAs (miRNAs) play an essential role in regulating plant development by mediating target genes at transcriptional and post-transcriptional levels [34], and DUF3537 (perimeter-associated SNP locus S06_39973848) is one of the predicted target genes of miRNAs in *Acacia crassicarpa* [35]. DUF3537 is a member of the transmembrane protein family. Transmembrane proteins are recognized to play a role in biological stress response [36]. Examples include pathogen-induced cysteine-rich transmembrane proteins [37], suppressors of NPR1 Constitutive2 proteins [36], polygalacturonase-inhibiting proteins [38,39,40], and ankyrin repeat-containing proteins [41].

The role of ribonucleotide reductase (perimeter and width: SNP loci S02_36533975 and S02_36673863) in the de novo synthesis of deoxynucleoside triphosphates (dNTPs) in DNA replication and cell cycle progression is critical [42]. Homeodomain leucine-zipper (perimeter: SNP locus S01_72472059) interacts genetically to align morphogenesis and environmental responses by modulating phytohormone-signaling networks [43]. SNP locus S02_46916695 (perimeter) is located next to Sobic.001G447400 and associated with TPR and ankyrin repeat. The TPR-containing protein TTL1in Arabidopsis regulates plant responses to abscisic acid (ABA) in seeds and seedlings [44]. Ankyrin repeat-containing proteins are essential in cell growth, development, and response to hormones and environmental stresses [45], and ankyrin-TPR repeats gene clusters in rice are associated with panicle branching diversity [46]. SNP locus S06_51902835, one of the top SNPs associated with perimeter, was 2571 bp away from calcium/calmodulin-dependent protein kinase. In rice, OsDMI3 (calcium/calmodulin-dependent protein kinase)-mediated phosphorylation of OsMKK1 (MAPK) kinase activates the MAPK cascade and positively regulates abscisic acid responses in seed germination, root growth, and tolerance to both water stress and oxidative stress [47]. Glycosyltransferase, tagged with seed perimeter-associated SNP locus S02_46926845, is known to have roles in seed coat mucilage composition [48]. Auxin is a plant hormone central to plant growth and development from embryogenesis to senescence, and PB1 domain (perimeter: SNP locus S04_67200818) interactions in auxin response factor ARF5 and repressor IAA17 [49]. The closest gene from the SNP locus S01_52374442 (width) was Sobic.001G271500, which contains a leucine-rich repeat. Leucine-rich repeat proteins are critical for growth promotion, seed maturation, stress response, and enhanced seed production [50,51]. Leucine-rich repeat proteins such as the polygalacturonase-inhibiting proteins are involved in plant defense [52]. The superfamily of cytochrome P450 tagged by SNP locus S03_5165375 (PCs) plays critical roles in plant growth and development, biotic and abiotic stress responses, and metabolic diversification [53,54]. Considering that PC1, which was used as input data, predominantly consists of variables such as area size, perimeter length, length, width, and other PCs included as covariates, the multivariate GWAS results mainly reflect traits associated with seed size.

The SNP locus that S06_58756099 (PCs) tagged to farnesyl-pyrophosphate synthetase is known to be associated with plant development [55]. The N-terminal domains of Arabidopsis rhamnose synthases RHM1, 2, and 3 have UDP-D-glucose 4,6-dehydratase activity (PCs-related SNP loci S04_62452658), and rhamnose synthases are required for the development of root hairs and cotyledon pavement cells and the synthesis of seed mucilage [56]. SNP loci S06_8316056 and S03_5151221 (PCs) were related to protein transport-related genes. In plants, the Golgi apparatus is central to synthesizing complex cell wall polysaccharides and glycolipids in the plasma membrane and adding oligosaccharides to proteins destined to reach the cell wall, plasma membrane, or storage vacuoles [57]. Rho GTPases (PCs-related SNP locus S04_62432641) modulate plant growth and development. Similarly, PPR proteins (PCs-related SNP locus S04_62438345) also play important roles in seed development, plant growth and development, and stress responses [58]. WD40 repeat genes, which include SNP loci S03_51995650 (PCs) and S10_15656073 (brightness), are reported to be associated with anthocyanin accumulation, seed pigmentation, seed germination, seed growth, and biomass [59,60]. DNA helicases, a gene close to SNP locus S01_53382065 associated with seed brightness, are molecular motor proteins that have suggested roles in cell division/proliferation during flower development, maintenance of genomic methylation patterns, and the plant cell cycle, as well as in basic cellular activities [61].

The brightness-related SNP locus S01_2657706 is found near amyloid beta precursor protein-binding protein 1 (APPBP1). Numerous amyloids are involved in pathogenesis; however, plant amyloids are poorly studied but are known to play roles in the autonomous flowering pathway and post-translational modification [62]. Ribonuclease III (brightness-related SNP locus S01_66050539) is responsible for the processing and maturation of RNA precursors into functional rRNA, mRNA, and other small RNA. However, no reported role of seed color or development has been known for the genes. SNP locus S06_37776989 (width) tagged gene of unknown function (DUF493) and SNP locus S10_53260179 (brightness) were closely located on a locus that does not have any annotated gene nearby. These SNPs may be associated with seed morphology but could also be false positives caused by pure coincidences.

Most of the genes identified are involved in plant development and physiological processes with potential aspects of seed morphology. Bonferroni correction is often considered highly conservative [63], and in this study, 100 SNPs passed the Bonferroni threshold. Furthermore, we verified the average score/SNP through a simple *t*-test and filtered out any SNP that failed to pass *p* < 0.05. Even with these strict methods to minimize false positives, there are many novel SNPs potentially conferring changes in various seed morphologies. Hence, it is expected that most identified SNPs are genuinely associated with seed morphology-related traits that can be used for sorghum breeding in the future. Further studies should investigate the relationship between seed morphology-related characteristics and molecular markers to better understand seed morphology-related genes in the subset of the Senegalese sorghum collection and other collections within the National Plant Germplasm System of the US.

## 4. Materials and Methods

### 4.1. Seed Phenotypic Evaluation

A total of 162 cultivars from the Senegalese germplasm collection (complete list available in Appendix A) maintained by the USDA-ARS, Plant Genetic Resources Conservation Unit, Griffin, Georgia, and Controls (BTx623, PI609251, and PI659985) were evaluated for seed area size (mm^2^), length (mm), width (mm), length-to-width ratio (LWR), perimeter (mm), circularity (0–1 range, 0: not circular to 1: complete circle), the distance between the intersection of length & width (IS) and center of gravity (CG), and seed darkness & brightness (0–255 range, 0: complete black to 255: pure white). CG is the point where the seed’s mass is concentrated and IS are points where the width and length hit the boundary for the seed parameter [64].

Around 80 to 100 seeds were evaluated for each trait across all the cultivars and controls, except for PI659985, where the available number of seeds was only 50. Seed images were captured with Canon imageRUNNER ADVANCE C7270 (Canon Inc., Tokyo, Japan). With the SmartGrain (version 1.3) high-throughput phenotyping software, the scanned seed images were measured for seed area size, length, width, LWR, perimeter, circularity, and distance between IS and CG [64]. Any errors generated by SmartGrain were manually corrected for each image. Seed darkness and brightness were measured using a multi-point function in ImageJ version 1.54d [65].

### 4.2. Statistical Analysis

Tukey’s HSD test for all possible cultivar comparisons was performed with JMP Pro 15 (SAS Institute, Cary, NC, USA) in each trait for statistical analysis. One-way ANOVA was performed for each trait separately. Pearson’s correlation was calculated for all possible pairs of seed morphology-related traits with JMP Pro 15. For anthracnose and head smut resistance traits [6,7,16,17], both Pearson’s (for parametric traits: all traits except anthracnose score-related traits) and Spearman’s rank (for non-parametric traits including the 1–5 scale scoring for anthracnose-related traits) correlation tests were performed. The Shapiro–Wilk test was conducted to evaluate normal distribution in each trait. A principal component analysis (PCA) was performed using data from the eight measured traits. Additional PCA was performed using seven correlated data, except the traits for color for multi-variate GWAS analysis.

### 4.3. GWAS and Population Genomic Analysis

For GWAS, a total of 193,727 SNP data were extracted from an integrated sorghum SNPs dataset based on sorghum reference genome version 3.1.1 and initially genotyped using GBS [18,66,67,68]. Phenotypic data was input univariately to perform GWAS with a mixed linear model (MLM) through TASSEL version 5.2.55 [69] association mapping software to identify chromosomal locations associated with each trait. Moreover, a PCA was performed with TASSEL version 5.2.55 for the traits that showed high correlations (all traits except seed color in this study), and a multivariate GWAS [PC GWAS based on PCs for phenotypic data (PC1 = data and other PCs = covariates)] was conducted through MLM. False associations were minimized by removing SNPs with higher than 20% unknown alleles and SNPs with minor allele frequency (MAF) below 5%, resulting in 132,024 SNPs [70]. The SNPs that passed the Bonferroni threshold were mapped back to the publicly available sorghum reference genome to be tracked to the specific chromosome location based on the sorghum reference genome sequence, version 3.1.1, available at the Phytozome 13 (https://phytozome.jgi.doe.gov, (accessed on 25 May 2023)) [71]. The mean values for Senegalese germplasms with either of the two prevalent bases were determined for each of the prospective genes listed in Table 3. The differences in these mean values were verified to be significant (*p* < 0.05) using JMP Pro 15 (SAS Institute, Cary, NC, USA). SNPs that did not pass the *t*-test were excluded from the list of candidate SNPs.

PLINK v1.9 [72] was used for VCF file conversions and to randomly select 50,000 SNPs from the genotypic data for analysis of population structure in R v4.1.2 and R studio v1.4.1717. The packages FactoMineR v2.8 [73] and Factoextra v1.0.7 [74] were used to conduct PCA and determine optimal k-means clustering using the average silhouette method, respectively. ADMIXTURE v1.3.0 [75] was run with the optimal k-means clustering value to visualize population structure. SNPRelate v1.28.0 and gdsfmt v1.30.lil0 [76] were used to generate a dendrogram of the accessions from the genotype data, which was visualized using ggtree v3.2.1 [73] to further validate population structure, and assign accessions to genetic groups. LD heatmap v1.0-6 [77] was utilized to plot LD of local variants around statistically significant SNPs.

## 5. Conclusions

In this study, we analyzed seed morphology-related traits in a subset of Senegalese germplasm. Even though a low genetic diversity was found, the accessions showed a wide range of morphological traits. Intriguingly, there were potential associations between seed morphology-related traits and head smut spot appearance rate, explained by possible genetic linkages. Nearly all the candidate genes from GWAS analysis had known roles in plant growth and development. The identified genes’ functions can be validated by using modern and cutting-edge techniques such as real-time quantitative reverse transcription PCR (Real-time qRT-PCR), RNA sequencing analysis (RNA-Seq), and CRISPR-Cas9-associated gene editing. Although applying gene-editing techniques in monocot crops is challenging, rapid developments in gene editing technology will offer fast and precise functional validations of the candidate genes. On the other hand, it is essential to survey seed morphology-related traits in other sorghum populations to identify additional candidate genes and genes that overlap in multiple populations.

## Figures and Tables

**Figure 1 plants-12-02344-f001:**
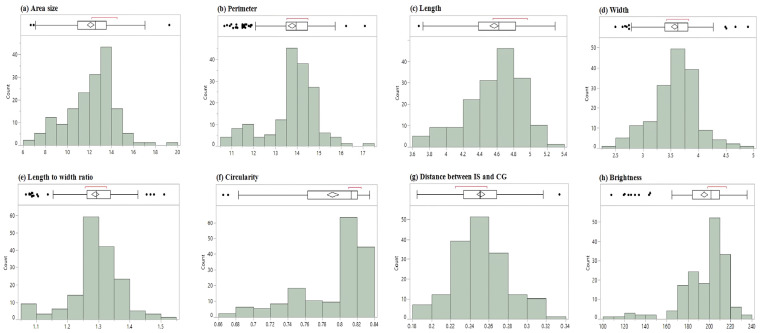
Histograms for distribution in Senegalese cultivars regarding seed morphologies. (**a**) Area size (mm^2^), (**b**) perimeter (mm), (**c**) length (mm), (**d**) width (mm), (**e**) length-to-width ratio, (**f**) circularity (0–1 range), (**g**) distance between IS and CG (mm) and (**h**) brightness (0–255 range). Box plots above each histogram indicated mean value (diamond), percentiles, upper and lower whisker, and outliers. Median ranges are shown in red.

**Figure 2 plants-12-02344-f002:**
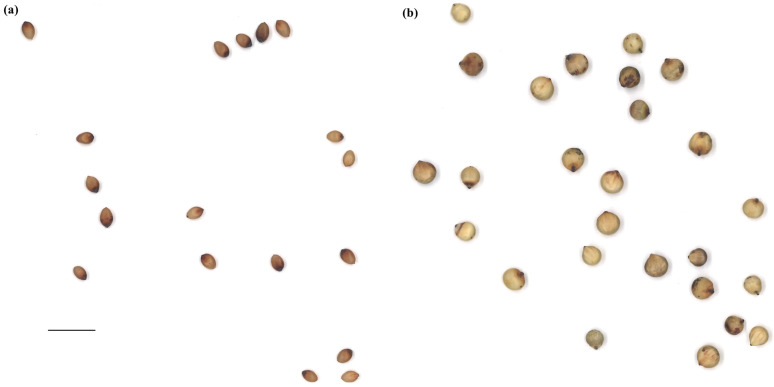
A comparison of the area sizes for PI514293 vs. PI514404. (**a**) PI514293 has one of the smallest seeds in area size. (**b**) PI514404 showed one of the largest seeds in the population. The scale bar indicates 1 cm applied to both (**a**,**b**).

**Figure 3 plants-12-02344-f003:**
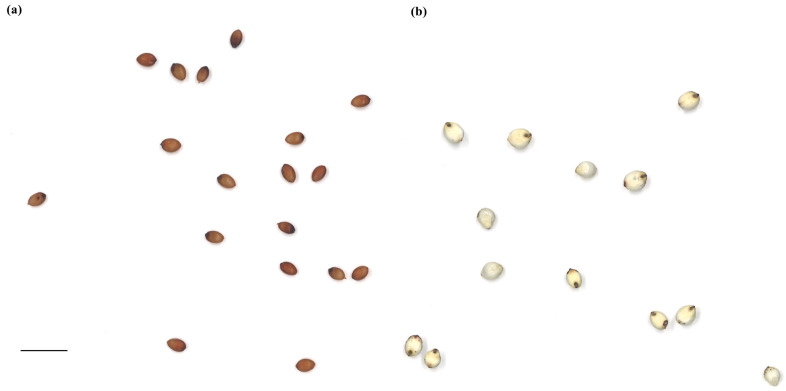
A comparison of the seed colors for PI514471 vs. PI514419. (**a**) PI514471 has one of the darkest seeds among the screened cultivars. (**b**) PI514471 showed one of the brightest seeds in the population. The scale bar indicates 1 cm applied to both (**a**,**b**).

**Figure 4 plants-12-02344-f004:**
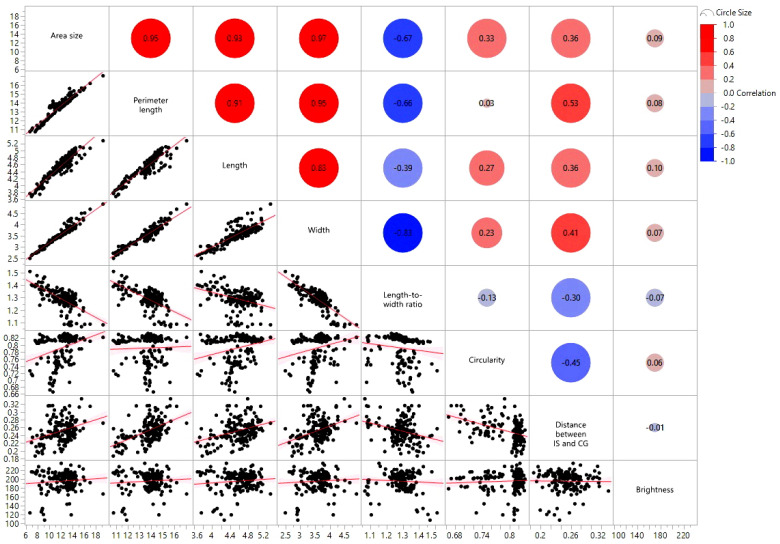
Scatter plots visualizing correlations between evaluated traits based on Pearson’s r. Each dot indicates a sorghum accession and red lines indicate correlations between two traits.

**Figure 5 plants-12-02344-f005:**
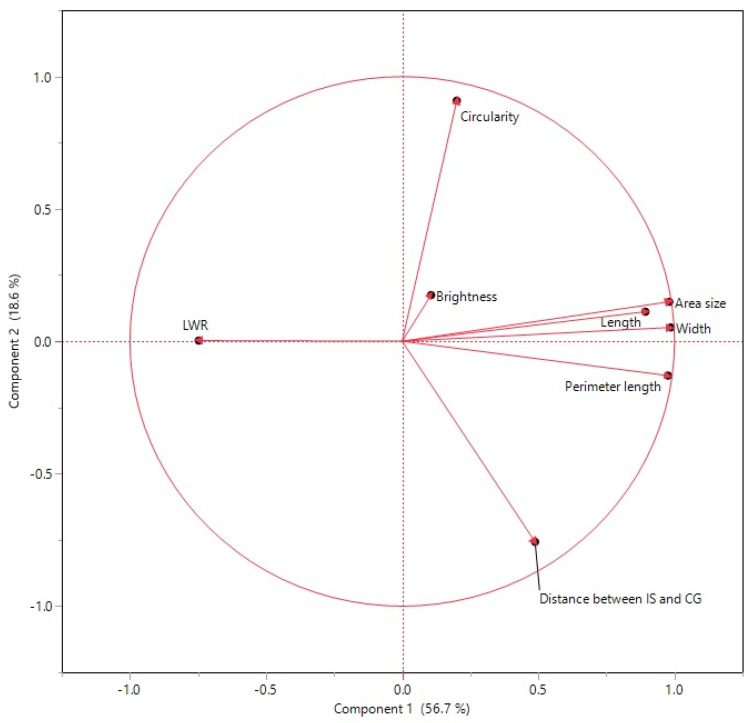
The principal component analysis of eight seed morphological parameters was from the Senegalese sorghum germplasms (162 Senegalese and 3 control cultivars). PC1 vs. PC2 are shown. Red arrows indicate directions for the traits on the PCA plot.

**Figure 6 plants-12-02344-f006:**
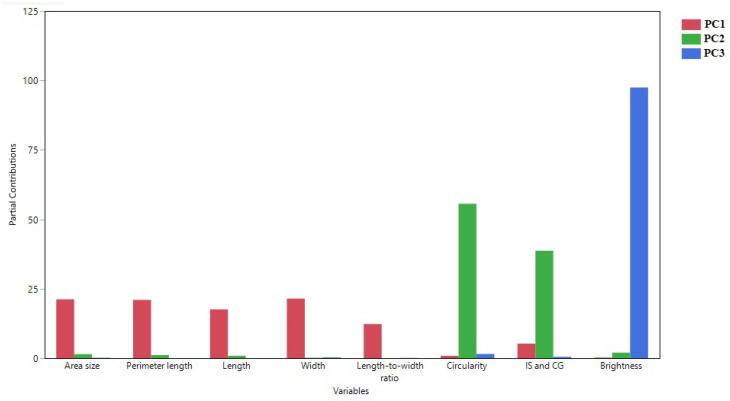
The plot of the partial contribution of variables for eight traits. Principal component analysis from Senegalese sorghum cultivars of eight seed morphological parameters. Contributions of each trait toward PC1, PC2, and PC3 are shown.

**Figure 7 plants-12-02344-f007:**
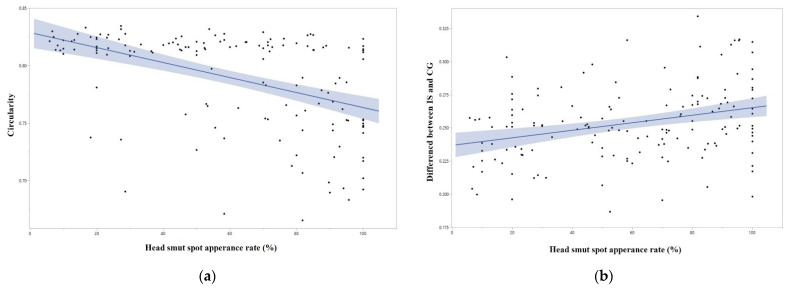
The plots of correlations between head smut spot appearance rate (%) and seed morphology-related traits. (**a**) Head smut spot appearance rate (%) and circularity showed a moderate negative correlation (Pearson’s correlation = −0.47, *p* < 0.0001). (**b**) Head smut spot appearance rate (%) and distance between IS and CG showed a moderate positive correlation (Pearson’s correlation = 0.31, *p* < 0.0001). Each dot indicates a sorghum accession; blue lines indicate correlations between the two traits.

**Figure 8 plants-12-02344-f008:**
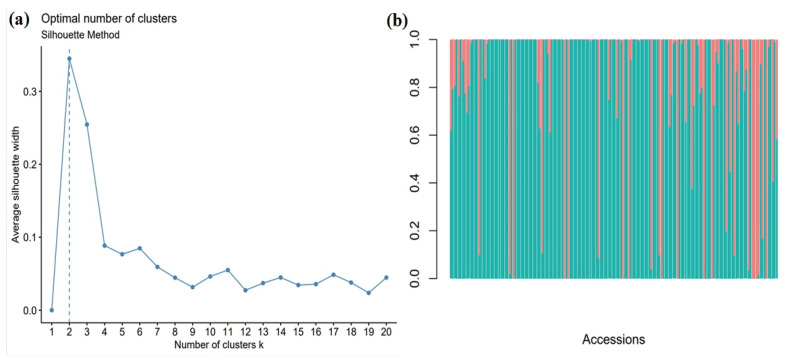
The population structure of the Senegalese sorghum panel. (**a**) k-means clustering by average silhouette method for genotype data of accessions analyzed in this study. The optimal number of k-means clusters was equal to 2, indicative of 2 major genetic groups. (**b**) An admixture plot displays 2 groups with most accessions falling in just one group. Two different colors indicate two different populations across the accessions.

**Figure 9 plants-12-02344-f009:**
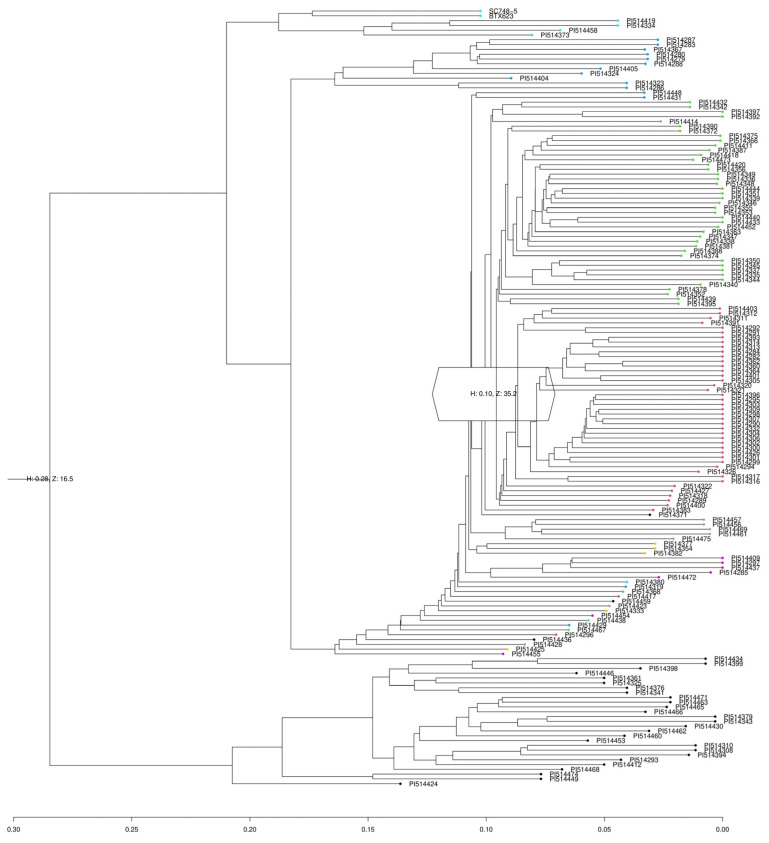
A dendrogram generated from genotype data of the accessions analyzed in this study. There are 2 major genetic groups, indicated by the two main branches, with that majority of the accessions falling in the top branch, which could potentially contain numerous subpopulations.

**Figure 10 plants-12-02344-f010:**
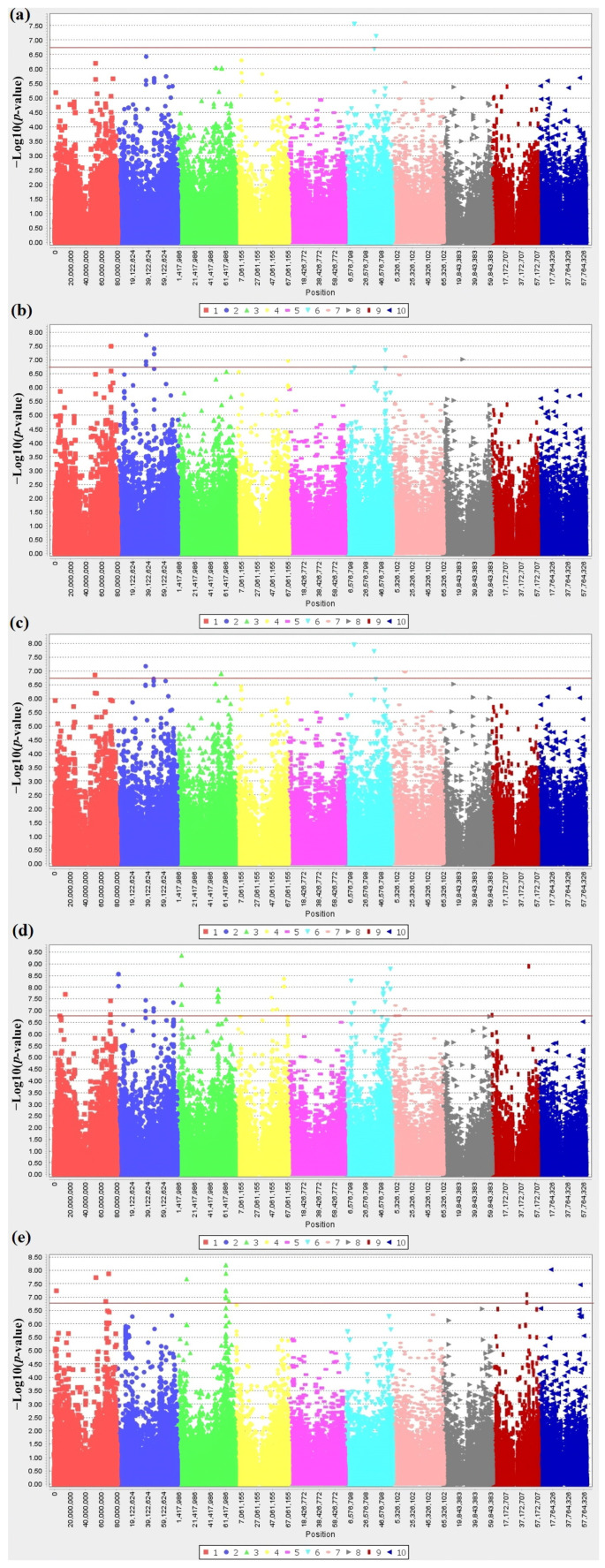
The genome-wide association for Senegalese sorghum seed morphology-related traits. Manhattan plots based on univariate and multivariate GWAS display the top candidate SNPs associated with each trait (**a**) Manhattan plot for area size. (**b**) Manhattan plot for perimeter (**c**) Manhattan plot for width. (**d**) Manhattan plot of PCs based on seven seed size-related traits. (**e**) Manhattan plot of seed brightness. The line is a cut-off for the Bonferroni threshold ≈ 0.00000017.

**Figure 11 plants-12-02344-f011:**
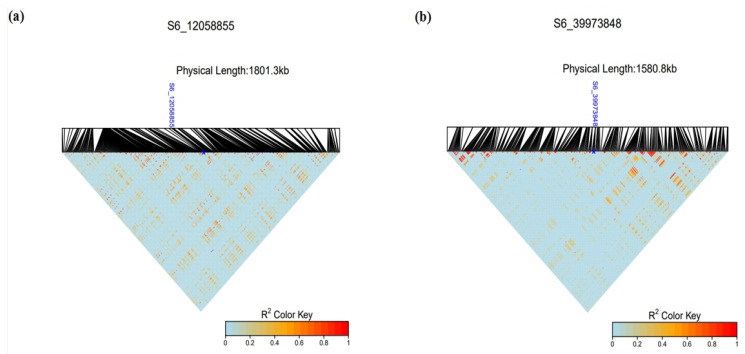
LD heatmaps visualize LD of regions around statistically significant SNP loci. (**a**) Variant S6_12058855 was associated with seed area. (**b**) Variant S6_39973848 was associated with seed perimeter. The blue stars indicate locations of the SNP loci.

**Table 1 plants-12-02344-t001:** The top five cultivars for each seed morphology are shown.

Largest Area Size (mm^2^)		Smallest Area Size (mm^2^)	
Accession	Mean ± S.D.	Accession	Mean ± S.D.
1. PI514404	19.26 ± 3.41	1. PI514293	6.67 ± 0.86
2. PI514287	17.03 ± 2.26	2. PI514308	6.94 ± 1.1
3. PI514367	16.10 ± 2.03	3. PI514310	7.11 ± 0.87
4. PI514464	15.88 ± 2.71	4. PI514388	7.56 ± 1.05
5. PI514458	15.84 ± 1.76	5. PI514474	7.71 ± 0.82
Longest perimeter (mm)		Shortest perimeter (mm)	
1. PI514404	17.09 ± 1.63	1. PI514388	10.78 ± 0.76
2. PI514287	16.23 ± 1.34	2. PI514293	10.79 ± 1.01
3. PI514367	15.73 ± 1.13	3. PI514474	10.92 ± 0.61
4. PI514288	15.7 ± 1.24	4. PI514308	10.99 ± 1.21
5. PI514283	15.65 ± 1.53	5. PI514449	11.07 ± 0.68
Longest length (mm)		Shortest length (mm)	
1. PI514404	5.29 ± 0.46	1. PI514308	3.67 ± 0.39
2. PI514429	5.11 ± 0.3	2. PI514310	3.72 ± 0.26
3. PI514464	5.09 ± 0.57	3. PI514434	3.74 ± 0.22
4. PI514458	5.09 ± 0.34	4. PI514293	3.76 ± 0.28
5. PI514478	5.08 ± 0.31	5. PI514388	3.78 ± 0.27
Longest width (mm)		Shortest width (mm)	
1. PI514404	4.9 ± 0.48	1. PI514293	2.49 ± 0.19
2. PI514287	4.67 ± 0.35	2. PI514308	2.62 ± 0.3
3. PI514367	4.51 ± 0.3	3. PI514310	2.67 ± 0.19
4. PI514288	4.49 ± 0.37	4. PI514388	2.72 ± 0.2
5. PI514458	4.27 ± 0.26	5. PI514325	2.74 ± 0.23
Highest LWR		Lowest LWR	
1. PI514293	1.51 ± 0.1	1. PI514288	1.07 ± 0.04
2. PI514325	1.48 ± 0.08	2. PI514283	1.08 ± 0.04
3. PI514471	1.47 ± 0.08	3. PI514285	1.08 ± 0.05
4. PI514462	1.45 ± 0.09	4. PI514404	1.08 ± 0.04
5. PI514446	1.43 ± 0.06	5. PI514367	1.09 ± 0.06
Highest circularity (0–1 scale)		Lowest circularity (0–1 scale)	
1. PI514425	0.83 ± 0.01	1. PI514282	0.67 ± 0.08
2. PI514440	0.83 ± 0.02	2. PI514292	0.67 ± 0.07
3. PI514420	0.83 ± 0.02	3. PI514294	0.68 ± 0.06
4. PI514434	0.83 ± 0.02	4. PI514297	0.69 ± 0.07
5. PI514409	0.83 ± 0.02	5. PI514296	0.69 ± 0.07
The longest distance between IS and CG (mm)		Shortest distance between IS and CG (mm)	
1. PI514288	0.33 ± 0.19	PI514388	0.19 ± 0.11
2. PI514294	0.32 ± 0.17	PI514398	0.19 ± 0.13
3. PI514292	0.32 ± 0.21	PI514446	0.2 ± 0.11
4. PI514283	0.32 ± 0.17	PI514438	0.2 ± 0.12
5. PI514341	0.32 ± 0.17	PI514449	0.2 ± 0.12
Brightest (0–255 scale)		Darkest (0–255 scale)	
1. PI514419	231.38 ± 19.43	1. PI514471	107.55 ± 10.76
2. PI514334	228.78 ± 15.28	2. PI514323	119.34 ± 20.39
3. PI514446	228.05 ± 18.19	3. PI514462	120.22 ± 12.47
4. PI514361	223.18 ± 15.75	4. PI514293	123.81 ± 20.84
5. PI514458	222.77 ± 20.39	5. PI514466	126.28 ± 15.52

**Table 2 plants-12-02344-t002:** Detailed correlations among eight seed morphology-related traits. *** = *p* < 0.0001, ** = *p* < 0.001 and * = *p* < 0.01.

	Area Size	Perimeter	Length	Width	LWR	Circularity	IS and CG	Brightness
Area size (mm^2^)	1 ***	0.95 ***	0.93 ***	0.97 ***	−0.67 ***	0.33 ***	0.36 ***	0.09
Perimeter (mm)	0.95 ***	1 ***	0.91 ***	0.95 ***	−0.66 ***	0.03	0.53 ***	0.08
Length (mm)	0.93 ***	0.91 ***	1 ***	0.83 ***	−0.39 ***	0.27 **	0.36 ***	0.10
Width (mm)	0.97 ***	0.95 ***	0.83 ***	1 ***	−0.83 ***	0.23 *	0.41 ***	0.07
LWR	−0.67 ***	−0.66 ***	−0.39 ***	−0.83 ***	1 ***	−0.13	−0.30 ***	−0.07
Circularity	0.33 ***	0.03	0.27 **	0.23 *	−0.13	1 ***	−0.45 ***	0.06
IS and CG	0.36 ***	0.53 ***	0.36 ***	0.41 ***	−0.30 ***	−0.45 ***	1 ***	−0.01
Brightness	0.09	0.08	0.10	0.07	−0.07	0.06	−0.01	1 ***

**Table 3 plants-12-02344-t003:** Annotated genes nearest to the most significant SNPs associated with GWAS regarding seed morphologies. SNPs that passed the Bonferroni threshold (0.00000017) were retested with a *t*-test for two prevalent bases, filtering out possible false positive SNPs. Multivariate GWAS based on PCs generated more than 50 SNPs passing Bonferroni correction (before filtering out with *t*-test), and the table shows only the top ten SNPs. Average score/SNP was not calculated for SNPs generated by PCs-based GWAS.

Trait	Chr	Location	Candidate Gene and Function	Distance (bp)	SNP Base %	TASSEL *p*-Value	Mean Value/SNP
Area	6	12,058,855 and one more SNP within 3 bp	Sobic.006G036400 Similar to OSJNBb0043H09.3 protein Ring zinc finger	136,397	A: 84% G: 16%	0.000000028	A: 12.65 G: 10.05
Perimeter	6	39,973,848	Sobic.006G056100 Protein of unknown function (DUF3,537)	0	A: 85.2% G: 14.8%	0.000000074	A: 14.11 G: 12.44
Perimeter	2	36,673,863	Sobic.002G147466 Ribonucleotide reductase-related	0	C: 92.9% G: 7.1%	0.000000013	C: 13.76 G: 14.7
Perimeter	1	72,472,059	Sobic.001G447400 Similar to homeobox-leucine zipper protein HAT22	2530	A: 16.3% C: 83.7%	0.000000033	A: 12.06 C: 14.14
Perimeter	2	46,916,695	Sobic.002G154750 Tetratricopeptide repeat-containing protein (TPR) Associated PlantFAMs- Ankyrin-1	6908	G: 16% T: 84%	0.00000004	G: 12.59 T: 14.09
Perimeter	6	51,902,835 and one more within 26 bp	Sobic.006G161000 Calcium/calmodulin-dependent protein kinase kinase (CAMKK)	2571	A: 15.2% G: 84.8%	0.000000043	A: 12.05 G: 14.13
Perimeter	2	46,926,845	Sobic.002G154800 Similar to glycosyl transferase family 1 protein-like Mannosylfructose-phosphate synthase/MFPS	0	C: 16.6% G: 83.4%	0.000000063	C: 12.38 G: 14.09
Perimeter	7	16,244,875	Sobic.007G093100 No annotation Associated PlantFAMs-zinc-finger of the FCS-type	37,632	C: 83.2% T: 16.8%	0.000000077	C: 14.13 T: 12.06
Perimeter	4	67,200,818	Sobic.004G340100 PB1 domain	2091	A: 15.9% G: 84.1%	0.00000011	A: 12.03 G: 14.13
Perimeter	2	36,482,136	Sobic.002G147433 Zinc finger	137,836	A: 6.4% G: 93.6%	0.00000012	A: 14.71 G: 13.74
Perimeter	2	36,533,975	Sobic.002G147466 Ribonucleotide reductase-related	139,787	C: 93.5% T: 6.5%	0.00000015	C: 13.74 T: 14.71
Width	6	12,058,855 and one more SNP within 3 bp	Sobic.006G036400 Similar to OSJNBb0043H09.3 protein Ring zinc finger	136,397	A: 84% G: 16%	0.000000011	A: 3.67 G: 3.2
Width	6	37,776,989	Sobic.006G050400 Protein of unknown function (DUF493)	0	A: 11.3% G: 88.7%	0.00000002	A: 2.99 G: 3.64
Width	2	36,673,863	Sobic.002G147466 Ribonucleotide reductase-related	0	C: 92.9% G: 7.1%	0.000000068	C: 3.56 G: 3.94
Width	7	16,244,901 and one more SNP within 26 bp	Sobic.007G093100 No annotation Associated PlantFAMs- zinc-finger of the FCS-type	37,606	A: 16.8% T: 83.2%	0.00000011	A: 3.05 T: 3.68
Width	1	52,374,442	Sobic.001G271500 Leucine-rich repeat-containing protein	22,206	G: 90.6% T: 9.4%	0.00000014	G: 3.63 T: 2.99
PCs	3	5,165,375	Sobic.003G058600 Cytochrome P450 CYP2 subfamily	0	A: 7.5% G: 92.5%	0.00000000044	
PCs	6	58,756,099	Sobic.006G248000 Farnesyl-pyrophosphate synthetase	0	C: 87.9% T: 12.1%	0.0000000017	
PCs	4	62,452,658 and one more SNP within 11 bp	Sobic.004G282700 Similar to DTDP-D-glucose 4,6-dehydratase-like	2043	G: 92.5 T: 7.5%	0.0000000044	
PCs	6	8,316,056	Sobic.006G033200 Similar to Probable protein transport Sec1a Syntaxin-binding protein 1 (STXBP1, MUNC18-1)	266,373	G: 25.3% T: 74.7%	0.0000000054	
PCs	6	54,692,844	Sobic.006G193600 Zinc-finger of the FCS-type, C2-C2 (zf-FLZ)	214	C: 94.2% G: 5.8%	0.0000000071	
PCs	6	54,708,681 and one more SNP within 11 bp	Sobic.006G193700 Zinc-finger of the FCS-type, C2-C2 (zf-FLZ)	0	G: 94.2% T: 5.8%	0.0000000071	
PCs	3	5,151,221	Sobic.003G058500 Golgi to er traffic protein 4 homolog	0	C: 92.5% T: 7.5%	0.0000000075	
PCs	4	62,432,641	Sobic.004G282400 Rho guanosine triphosphatases (GTPase) activating protein 2	0	C: 93.1 T: 6.9%	0.0000000095	
PCs	4	62,438,345	Sobic.004G282500 Pentatricopeptide (PPR) repeat-containing protein-like	11	C: 93.1 T: 6.9%	0.0000000095	
PCs	3	51,995,650	Sobic.003G195900 Stomatal cytokinesis defective Associated PlantFAMs- WD40 repeat-containing protein	0	A: 5.7% G: 94.3	0.000000012	
Brightness	10	15,656,073	Sobic.010G126000 WD-40 repeat protein	17,962	C: 92.5% T: 7.5%	0.0000000093	C: 198.62 T: 156.23
Brightness	1	53,382,065	Sobic.001G275300 ATP-dependent DNA helicase	11,647	G: 7.7% T: 92.3%	0.000000019	G: 186.61 T: 198.24
Brightness	10	53,260,179	Sobic.010G191301 No annotation Highly expressed in panicle-upper anthesis	7571	C: 7% G: 93%	0.000000035	C: 177.47 G: 197.87
Brightness	1	2,657,706	Sobic.001G035200 Amyloid beta precursor protein-binding protein 1 (APPBP1)	0	A: 92.4% G: 7.6%	0.000000058	A: 198.38 G: 156.23
Brightness	1	66,050,539	Sobic.001G372400 No annotation Associated PlantFAMs- Ribonuclease III	806	C: 94.3% T: 5.7%	0.00000015	C: 198.3 T: 155.85

## Data Availability

Not applicable.

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
