# Peer review of "Genome-Wide Association Study of Seed Morphology Traits in Senegalese Sorghum Cultivars"

_plants, 2023, doi:10.3390/plants12122344_

Round 1

Reviewer 1 Report

The study presents a very  important area of Sorghum breeding and is well planned, executed with appropriate methodology supported with statistical analysis. The results of the study were also well presented supported by relevant discussion. Only two minor revisions are to be made before acceptance of the manuscript for publication:

(1) At few places where the references need to be cited as per the format of the journal.

(2) Conclusion to be given at the end of the manuscript after discussion.

Author Response

We appreciate the reviewers' efforts to improve the manuscript. The review was thorough, resulting in great improvement! Here are our replies to the reviewer’s comment.

(1) At few places where the references need to be cited as per the format of the journal. We revised the issues except for a few places shown for product information and company name that were not meant to cite.

(2) Conclusion to be given at the end of the manuscript after discussion. We added the conclusion part as suggested.

Reviewer 2 Report

The article “Genome-wide association study of seed morphology traits in Senegalese sorghum cultivars” presented significant SNPs and mapped to the reference sorghum genome to uncover multiple candidate genes potentially associated with seed morphology. However, there are still shortcomings which must be revised.

Mention in the abstract from where the data was collected.

Also add some main results of the study.

Results of genome wide association should be included.

Main methods should be added in the abstract.

Use the abbreviation at first use.

Line 40 needs to be cited with more recent studies such as https://doi.org/10.3390/agronomy13010269, https://doi.org/10.1080/01916122.2022.2144521

Line 47 should be cited with recent study https://doi.org/10.1016/j.cclet.2021.04.022

In the case of seed morphology traits, there may be environmental factors that influence the phenotype, such as soil type, temperature, and water availability. It is important to account for these factors in the analysis to avoid false positive associations.

Secondly, the choice of sorghum cultivars is also important. Senegal has a diverse range of sorghum cultivars, each with unique genetic backgrounds and phenotypic traits. It is important to choose cultivars that are representative of the population and have sufficient genetic diversity to capture the full range of variation in the trait of interest. So provide sufficient information of the selected cultivars in the introduction section.

GWAS identifies associations between genetic variants and traits, but it does not necessarily imply causality. Further functional studies are needed to confirm the biological relevance of the identified variants and their potential role in determining seed morphology traits.

Figure 2 and 3 are not clear.

Add conclusion with future perspective and research gaps using the current results.

revise long sentences to convey a clear message to readers

Author Response

We appreciate the reviewers' efforts to improve the manuscript. The review was thorough, resulting in great improvement! Here are our replies to the reviewer’s comment.

Comments and Suggestions for Authors

Comments:

Comments and Suggestions for Authors

The article “Genome-wide association study of seed morphology traits in Senegalese sorghum cultivars” presented significant SNPs and mapped to the reference sorghum genome to uncover multiple candidate genes potentially associated with seed morphology. However, there are still shortcomings which must be revised.

Mention in the abstract from where the data was collected. A correction was made as suggested.

Also add some main results of the study. A correction was made as suggested.

Results of genome wide association should be included. A correction was made as suggested.

Main methods should be added in the abstract. A correction was made as suggested.

Use the abbreviation at first use. A correction was made as suggested.

Line 40 needs to be cited with more recent studies such as https://doi.org/10.3390/agronomy13010269, https://doi.org/10.1080/01916122.2022.2144521

Line 47 should be cited with recent study https://doi.org/10.1016/j.cclet.2021.04.022

We checked the literature, and they look fantastic. Unfortunately, we don’t understand the information from the studies as it is beyond our expertise.

In the case of seed morphology traits, there may be environmental factors that influence the phenotype, such as soil type, temperature, and water availability. It is important to account for these factors in the analysis to avoid false positive associations.- That is true, but this study provides good information regarding seed morphology as plants were grown in the exact location.

Secondly, the choice of sorghum cultivars is also important. Senegal has a diverse range of sorghum cultivars, each with unique genetic backgrounds and phenotypic traits. It is important to choose cultivars that are representative of the population and have sufficient genetic diversity to capture the full range of variation in the trait of interest. So provide sufficient information of the selected cultivars in the introduction section.- We mentioned that we chose these cultivars as they are unexplored for the traits. We are currently working on another population of sorghum for various traits.

GWAS identifies associations between genetic variants and traits, but it does not necessarily imply causality. Further functional studies are needed to confirm the biological relevance of the identified variants and their potential role in determining seed morphology traits.- It is true that functional studies are needed. As this manuscript include lots of information, we are planning to do functional studies for the next manuscript.

Figure 2 and 3 are not clear.- It must be due to file conversion. We uploaded the jpeg files to “Plants”

Add conclusion with future perspective and research gaps using the current results. A correction was made as suggested.

Reviewer 3 Report

The authors describe in their study the results of a genome-wide association study on seed morphology traits in sorghum germplasm. Based on the associated SNPs they were able to identify several candidate genes for the morphology traits. Additionally, they checked for correlation between seed morphology and resistance towards two fungal diseases and found two significant correlations. The article is for the most part scientifically sound and most comments just refer to minor errors.

Comments:

Abstract:

- Line 17-18: Stating that the seed morphology traits and anthracnose and head smut resistance traits were correlated is misleading. No significant correlation regarding anthracnose is reported in the paper. "Correlations between seed morphology-related traits and traits associated with anthracnose and head smut resistance were analyzed." may be a more suitable wording.

Introduction:

- Line 45: Please clarify in which regard the seed shape and size are of particular importance.

- Line 67-69: Same problem as in the abstract.

Materials and methods:

- Line 81: IS and CG have already been defined in the previous paragraph and don't need to be written out again here. Alternatively, you can write out LWR in Line 79 as well.

- Line 98-99: Please clarify what is meant by "parametric" and "non-parametric" traits.

- Line 98-99: At what point was spearman rank correlation used instead of pearson? All reported correlation values are indicated to be based on pearson correlation. Please clarify for what you used spearman.

- Line 100-101: You state that the PCA was done using only seven of the eight traits. However, in Line 162 you write that the PCA was done using eight traits.

- Line 109-110: Clarify whether you mean by "PC1=data" that you use PC1 as the phenotype for the GWAS.

- Line 110-112: Add the number of SNPs that were removed using these filtering steps.

- Line 112: Add the actual value of the Bonferroni threshold.

- Line 117: The prospective genes are listed in Table 3 and not Table 1.

- Line 121: The version of RStudio that has been used is practically irrelevant. Instead add the version R that you used.

Results:

- Line 140: "PI514404" instead of "514404"

- Line 230: This should be "Table 3" instead of "Table 2".

- Figure 1: Add the meaning of the red interval above the boxplot to the figure caption.

- Table 1: Remove the additional dot at the end of the caption.

- Figure 4: If possible increase the contrast between circle color and numbers.

- Figure 6: Change the legend labels from "Prin*" to "PC*" so that it matches the figure caption.

- Figure 7: Increase the axis text size.

- Table 3:

- Use thousand separator for column "Base pairs" uniformly (e.g. right now there are 6,908 as well as 7517)

- Clarify what is meant by the column "Base pairs" and add the meaning to the caption. If the distance between SNP and gene transcript is meant, I would suggest to use "Distance (bp)" as column header.

- Clarify what is meant by the column "Average score/SNP" and add the meaning to the caption. If these are the mean values mentioned in Lines 115-117, then a column name like "Mean value/SNP" seems more suitable.

- There seems to be no discernible order for the table rows expect by "Trait". For the same trait, rows of multiple different chromosomes are mixed up. Please order the rows by "Trait" followed by "Chr" followed by "Location".

Discussion:

- Line 273: The reference for "Ahn et al" is missing.

Supplementary S1:

- What is the difference between the tables "Raw_data_Senegal_all_seed" and "Senegal_all_seed_size_alldata"? They seem to contain the same data, which makes one of them superfluous.

Author Response

We appreciate the reviewers' efforts to improve the manuscript. The review was thorough, resulting in great improvement! Here are our replies to the reviewer’s comment.

Comments and Suggestions for Authors

Comments:

Abstract:

- Line 17-18: Stating that the seed morphology traits and anthracnose and head smut resistance traits were correlated is misleading. No significant correlation regarding anthracnose is reported in the paper. "Correlations between seed morphology-related traits and traits associated with anthracnose and head smut resistance were analyzed." may be a more suitable wording.-Absolutely right, and we made a correction accordingly.

Introduction:

- Line 45: Please clarify in which regard the seed shape and size are of particular importance.-A correction was made as suggested.

- Line 67-69: Same problem as in the abstract. A correction was made as suggested.

Materials and methods:

- Line 81: IS and CG have already been defined in the previous paragraph and don't need to be written out again here. Alternatively, you can write out LWR in Line 79 as well.- Correction was made in line 79, but the explanation of IS and CG was not removed as one of the co-authors thinks the explanation is needed.

- Line 98-99: Please clarify what is meant by "parametric" and "non-parametric" traits.- Clarification was made.

- Line 98-99: At what point was spearman rank correlation used instead of pearson? All reported correlation values are indicated to be based on pearson correlation. Please clarify for what you used spearman.- Anthracnose-related traits (1-5 scale) were the only traits that we used Spearman’s, resulting in no significance.

- Line 100-101: You state that the PCA was done using only seven of the eight traits. However, in Line 162 you write that the PCA was done using eight traits.- We apologize for the confusion. Seven traits were done for PCA, entering for multivariate-GWAS analysis that is correlated. Separate PCA was done for all traits to find out the distance between all eight traits. We made a correction accordingly.

- Line 109-110: Clarify whether you mean by "PC1=data" that you use PC1 as the phenotype for the GWAS.- All PCs are phenotypic data, but the emphasis is on PC1. We added the word that PCs are phenotypic data.

- Line 110-112: Add the number of SNPs that were removed using these filtering steps.- We added the number of SNPs after filtering as recommended.

- Line 112: Add the actual value of the Bonferroni threshold. It is listed in Figure 10 (Bonferroni threshold≈ 0.00000017).

- Line 117: The prospective genes are listed in Table 3 and not Table 1. A correction was made accordingly.

- Line 121: The version of RStudio that has been used is practically irrelevant. Instead add the version R that you used.- We revised the part as suggested.

Results:

- Line 140: "PI514404" instead of "514404" A correction was made accordingly.

- Line 230: This should be "Table 3" instead of "Table 2". A correction was made accordingly.

- Figure 1: Add the meaning of the red interval above the boxplot to the figure caption. .- We revised the part as suggested.

- Table 1: Remove the additional dot at the end of the caption.- We revised it as suggested.

- Figure 4: If possible increase the contrast between circle color and numbers. .- We revised the part as suggested.

- Figure 6: Change the legend labels from "Prin*" to "PC*" so that it matches the figure caption. - We revised it as suggested.

- Figure 7: Increase the axis text size. - We revised the part as suggested.

- Table 3:

- Use thousand separator for column "Base pairs" uniformly (e.g. right now there are 6,908 as well as 7517) - Correction was made as suggested.

- Clarify what is meant by the column "Base pairs" and add the meaning to the caption. If the distance between SNP and gene transcript is meant, I would suggest to use "Distance (bp)" as column header.- Yes, it means distance (bp). We made a correction as suggested.

- Clarify what is meant by the column "Average score/SNP" and add the meaning to the caption. If these are the mean values mentioned in Lines 115-117, then a column name like "Mean value/SNP" seems more suitable.- Yes, it is Mean value/SNP. We made a correction as suggested.

- There seems to be no discernible order for the table rows expect by "Trait". For the same trait, rows of multiple different chromosomes are mixed up. Please order the rows by "Trait" followed by "Chr" followed by "Location".- It is ordered with traits first, and then p-values.

Discussion:

- Line 273: The reference for "Ahn et al" is missing- Correction was made as suggested.

Supplementary S1:

- What is the difference between the tables "Raw_data_Senegal_all_seed" and "Senegal_all_seed_size_alldata"? Yes, they were truly identical. We replaced one of them with average values for each accession.